# HER2 Inhibition in Gastric Cancer—Novel Therapeutic Approaches for an Established Target

**DOI:** 10.3390/cancers14153824

**Published:** 2022-08-06

**Authors:** Caroline Fong, Ian Chau

**Affiliations:** Gastrointestinal/Lymphoma Unit, The Royal Marsden NHS Foundation Trust, London SW3 6JJ, UK

**Keywords:** gastric cancer, HER2, immunotherapy, trastuzumab, antibody-drug conjugate, trastuzumab deruxtecan

## Abstract

**Simple Summary:**

The human epidermal growth factor receptor 2 (HER2) became the first routinely targeted biomarker in the management of stomach cancers when the monoclonal antibody trastuzumab given with chemotherapy was shown to improve patient survival in 2010. Over the past decade, we have developed our understanding of how HER2-directed drugs lose their effectiveness and have now reached a phase where several new drugs are being developed simultaneously for HER2-positive stomach cancers. In this paper, we will summarise the evidence supporting HER2 targeting in stomach cancers, review mechanisms of drug resistance and outline the new treatment approaches that may change the way we treat this disease.

**Abstract:**

Gastric cancer is a leading cause of cancer-related deaths globally. Human epidermal growth receptor 2 (HER2) overexpression of *HER2* gene amplification is present in 20% of gastric cancers and defines a subset amenable to HER2-directed therapeutics. The seminal ToGA study led to routine use of the monoclonal antibody trastuzumab in conjunction to platinum-fluoropyridimine first-line chemotherapy for HER2-positive gastric cancers as standard-of-care. Although limited progress was made in the decade following ToGA, there is now an abundance of novel therapeutic approaches undergoing investigation in parallel. Additionally, new data from randomised trials have indicated efficacy of the antibody-drug conjugate trastuzumab deruxtecan in chemorefractory patients and increased responses with the addition of first-line immune checkpoint blockade to trastuzumab and chemotherapy. This review will outline the data supporting HER2 targeting in gastric cancers, discuss mechanisms of response and resistance to HER2-directed therapies and summarise the emerging therapies under clinical evaluation that may evolve the way we manage this subset of gastric cancers in the future.

## 1. Introduction

The transmembrane receptor erbB-2, otherwise known has the human epidermal growth receptor 2 (HER2), is a member of the epidermal growth factor family of receptor tyrosine kinases. *HER2* is located on the human chromosome 17 (17q21) and encodes the transmembrane glycoprotein p185 [1,2]. The HER2 receptor consists of extracellular, transmembrane and intracellular tyrosine kinase domains [3]. HER2 overexpression is usually caused by gene amplification, and subsequent heterodimerisation with other members of the EGFR family precipitates tumour cell proliferation, invasion and metastasis predominantly through the RAS/MAPK and PI3K/Akt pathways. Tumours with HER2 overexpression and/or *HER2* amplification represent a molecularly defined subgroup of malignancies, which are observed in breast, gastric, colorectal, biliary tract and lung cancers. Therapeutic exploitation of HER2 has largely been led by advances seen in breast and gastric cancers, although its role in other solid tumour types is being actively explored [4].

Gastric adenocarcinoma is an aggressive cancer associated with poor prognosis. On a global scale, only 5–10% of patients who present with metastatic disease survive beyond five years of initial diagnosis [5]. Approximately 20% of gastric cancers are HER2-positive and are more common in tumours with intestinal pathology and junctional origin [6]. Although its prognostic role in gastric cancers is unclear, HER2 was the first routinely targeted biomarker in gastric cancers and, until the relatively recent advent of immunotherapy in gastrointestinal cancers, the only biomarker with implications on the clinical management of this disease. This review will summarise the current standard-of-care therapy for HER2-positive gastric cancers and examine novel approaches in the rapidly evolving therapeutic landscape for these cancers.

## 2. Trastuzumab as First-Line Therapy for HER2-Positive Advanced Gastric Cancers

The monoclonal antibody trastuzumab inhibits HER2-mediated signalling by preventing heterodimerisation, receptor internalisation and degradation, inhibition of the PI3K-AKT signalling pathway and antibody-dependent cellular cytotoxicity (ADCC). The clinical benefit of trastuzumab in advanced gastric cancers was first demonstrated in the ToGA study, which randomised 594 patients with HER2-positive gastric or gastro-oesophageal junction (GOJ) adenocarcinoma to trastuzumab and platinum-fluoropyrimidine chemotherapy or chemotherapy alone [7]. Eligible patients were considered HER2-positive if tumours were identified by HER2 immunohistochemical (IHC) staining of 3+ (defined as moderate to strong complete or basolateral membrane reactivity in >10% of tumour cells), or if *HER2* gene amplification by fluorescence in-situ hybridisation (ISH) (defined as HER2/centromeric probe for chromosome 17 (CEP17) ratio of >2) was deemed HER2-positive. In the overall study population, the addition of trastuzumab to chemotherapy improved median overall survival (OS) from 11.1 months with chemotherapy alone to 13.8 months (hazard ratio (HR) 0.74; 95% confidence interval (95% CI) 0.60–0.91; *p* = 0.0046). Post-hoc exploratory analysis showed that the most pronounced survival benefit from the addition of trastuzumab was seen in patients with IHC 2+/FISH positive or IHC 3+ tumours (median OS: 16.0 vs. 11.8 months; HR 0.65 (95% CI 051–0.83)), suggesting that higher levels of HER2 overexpression, perhaps more so than amplification, are a prerequisite for trastuzumab efficacy [7]. Based on these findings, trastuzumab in combination with platinum-fluoropyrimidine chemotherapy was established as the standard-of-care first-line treatment in patients with HER2 IHC 2+/ISH positive or IHC 3+ tumours.

## 3. Molecular Heterogeneity in Gastric Cancer: The Achilles’ Heel of HER2-Targeting

Progress in HER2 targeting in OG adenocarcinoma has largely stagnated following the initial success seen with ToGA. Therapeutic strategies such as dual inhibition with trastuzumab and pertuzumab or second-line therapy with the antibody-drug conjugate trastuzumab emtansine reported negative results [8,9] (Table 1), which are incongruous to the benefit seen with these approaches in HER2-positive breast cancer. The addition of lapatinib to first- and second-line chemotherapy also did not improve survival in gastric cancers [10,11] (Table 1).

Gastric cancer is a notoriously heterogenous disease where genomic and phenotypic disparities exist within individual tumours and between primary and metastatic sites. It is this molecular diversity that may have impeded the development of targeted therapeutics in this tumour type [15]. In gastric cancers, HER2 expression using IHC demonstrates a more heterogenous basolateral membrane staining pattern in comparison to breast cancers, with one series reporting heterogenous expression in 79% of tumours [16]. This differential staining pattern is reflected in the HER2 scoring system developed specifically for gastric cancers [17]. Additionally, HER2 testing guidelines jointly issued by the College of American Pathologists, American Society of Clinical Pathology and the American Society of Clinical Oncology advocate that a minimum of five biopsy specimens should be obtained to account for intratumoural heterogeneity in HER2 expression [18]. From a clinical perspective, inconsistent HER2 expression has implications on treatment efficacy, with heterogenous expression within the primary tumour being associated with shorter progression-free survival (PFS) on first-line trastuzumab-containing regimens relative to patients with homogenous expression [19,20].

To compound the issue of intra- and inter-tumoural molecular heterogeneity, prospective data from the VARIANZ study, which recruited patients from 35 centres throughout Germany to compare local and central HER2 assessment, also showed that interobserver variability can also account for discrepancies in HER2 IHC expression [21]. From a total of 548 patients recruited, 14.1% (n = 77) were deemed HER2-positive on central assessment. A deviation rate of 22.7% (n = 83) between central and local results was seen. The majority of discrepant cases were due to locally assessed HER2-positive cases being evaluated as HER2-negative centrally (n = 74/83). Notably, patients with HER2-positive disease on local assessment who were subsequently confirmed as HER2-positive centrally (n = 60) had significantly better mOS compared to those who were HER2-negative on central review (n = 65) (mOS: 20.5 vs. 10.9 months; HR 0.42; *p* < 0.001). The subset of patients with centrally confirmed HER2-positive status had significantly more tumour cells staining as HER2-positive and a higher *HER2* amplification ratio compared to those with unconfirmed HER2-positive status. The threshold of ≥40% HER2-positive cells and a HER2/CEP17 ratio of 3.0 were found to optimally select for benefit from trastuzumab in this cohort of patients. These are higher than the thresholds of >10% that are used to define HER2 3+ using IHC and HER2/CEP17 ratio of 2 in IHC 2+ patients, and which are conventionally used to define HER2-positivity in gastric cancers [17].

The GASTHER1 study also demonstrated that patients with HER2-positive disease can be identified on repeat biopsy of primary tumours or recurrent or metastatic disease on patients whose tumours were previously established as HER2-negative [22]. When treated with trastuzumab-containing first-line systemic therapy regimens, these patients had a PFS comparable to patients with HER2-positive disease at the outset. These results illustrate that re-biopsies could be considered on the development of disease relapse or progression in HER2-negative tumours, as it could dictate eligibility for HER2-targeted therapies. Conversely, downregulation of HER2 expression is also a recognised mechanism of acquired resistance to HER2-directed therapies. Loss of HER2 expression on paired biopsies is seen in approximately 30% of patients following trastuzumab-containing treatment [23,24], with no objective responses to second-line trastuzumab emtansine seen in patients with HER2-loss [24]. Progression-free survival was also comparatively shorter in this group of patients compared to those with sustained HER2 expression [24]. It should be noted that HER2 testing to determine patient eligibility for second-line randomised trials that reported negative results, such as TyTAN and GATSBY, was conducted on archival tissue that was collected prior to first-line therapy [9,11]. Other mechanisms of acquired resistance to HER2-targeted therapies include alterations in the HER2 receptor secondary to aminotruncation, resulting in the loss of the trastuzumab binding region, *FGFR* amplification and activation of the *MAP/ERK* and *PI3K/mTOR* downstream pathways [25,26,27]. Additionally, there is evidence that micro RNAs (miRNAs), non-coding RNAs that regulate target genes at the post-transcriptional level, may underpin trastuzumab resistance, and that serum-based miRNA signatures can distinguish trastuzumab-sensitive from resistant patients [28,29,30,31].

However, repeated tumour biopsies to reassess HER2 expression or investigate mechanisms of secondary resistance are unpleasant from the patient perspective, and each individual procedure carries a risk profile that may be detrimental to patient safety. The use of circulating tumour DNA (ctDNA) has the potential to not only minimise the risk and discomfort associated with serial biopsies but provide insight into targetable genomic alterations and mechanisms of resistance secondary to therapeutic pressure. Discordance in genomic aberrations between primary and metastatic tumours in oesophago-gastric adenocarcinomas have been documented, with one study reporting discrepancy in 36% (n = 10/28) paired primary and metastatic samples, which resulted in treatment reassignment in 9 patients [32]. In comparison, concordance of targetable alterations between metastatic tissue and cell-free DNA was higher (87.5%), suggesting that liquid biopsies have the potential to inform therapy selection in gastric cancers. High concordance of *HER2* amplification between baseline ctDNA and tumour tissue has been demonstrated, and the co-presence of baseline *PIK3CA/R1/C3* or *ERBB2/4* mutations in ctDNA were associated with poor PFS [33]. Longitudinal ctDNA monitoring has also identified *NF1* mutations to be indicative of trastuzumab resistance; similarly, ctDNA-detectable MET amplifications underlie acquired resistance to the tyrosine kinase inhibitor afatinib in patients with HER2-positive gastric cancers [34]. Collectively, these features exemplify the challenges that spatial and temporal heterogeneity pose to the progress of HER2-targeted therapies and highlight the need for innovative approaches to future drug and clinical trial design.

## 4. Novel Therapies for HER2-Positive Gastric Cancers

### 4.1. Antibody-Drug Conjugates

#### 4.1.1. Trastuzumab Deruxtecan

Trastuzumab deruxtecan (T-DXd or DS-8201a) is an antibody-drug conjugate (ADC) which consists of a humanised, monoclonal, anti-HER2 antibody bound to topoisomerase I inhibitor via a peptide-based linker [35]. Although the linker is stable in plasma, it is cleaved by lysosomal enzymes following internalisation into cancer cells. T-DXd has several novel features. It has a drug-to-antibody ratio of 8:1, which results in a higher drug payload than trastuzumab emtansine. The topoisomerase I payload of T-DXd is approximately 10 times more potent than SN-38, the active metabolite of irinotecan. The drug payload is also membrane permeable and can therefore diffuse into neighbouring cells, precipitating a bystander killing effect [36]. Overall, these characteristics facilitate efficient delivery of a potent cytotoxic payload to HER2-positive cells capable of overcoming heterogenous HER2 expression and simultaneously limiting off-target toxicity.

Clinical efficacy of T-DXd in HER2-positive advanced gastric adenocarcinoma patients was first detected in a phase Ib study, where 44 patients were treated with either 5.4 mg/kg or 6.4 mg/kg of T-DXd [37]. An objective response was observed in 43.2% of patients (n = 19/44) and disease control was achieved in 79.5% (n = 35/44). The median progression-free survival (PFS) was 5.6 months and median OS was 12.8 months. Based on the combined safety and efficacy data seen in this study, 6.4 mg/kg was selected as the dose for gastric and GOJ adenocarcinoma patients.

Building on these results, DESTINY-Gastric01 was a randomised phase II study within a chemorefractory Asian population, which randomised 187 HER2-positive patients in a 2:1 ratio to receive T-DXd or investigator’s choice chemotherapy (irinotecan or paclitaxel) [38]. Eligible patients had received at least two prior lines of therapy, including trastuzumab, and HER2 status was locally determined using patients’ most recent archival tissue. The objective response rate (ORR) by independent central review was 51% in patients who received T-DXd, which was significantly higher than the ORR of 14% seen in with chemotherapy (*p* < 0.001). Confirmed ORR was also higher with T-DXd compared to chemotherapy (43% vs. 12%), and 10 patients treated with T-DXd achieved complete responses. As the primary endpoint was met, the OS was statistically evaluated. Median OS with T-DXd was significantly longer than that seen with chemotherapy (12.5 vs. 8.4 months; HR 0.59; *p* = 0.001). Median PFS was also longer in T-DXd patients (5.6 vs. 3.5 months; HR 0.47). Common adverse events related to T-DXd such as cytopenias, anorexia and nausea were managed with dose modifications and supportive measures. Cardiotoxic effects, including reduction in left ventricular ejection fraction, which is a recognised side-effect of trastuzumab, were not observed with T-DXd. Of note, 12 patients in the T-DXd cohort (10%) had drug-related interstitial lung disease, as determined by an independent adjudication committee. Two cases assessed as grade 3 and one case as grade 4 and no fatal drug-induced ILD events occurred. These pivotal results precipitated Food and Drug Administration approval for the use of T-DXd in HER2-positive patients who have received a trastuzumab-containing regimen in addition to approval in Japan and is the first drug to improve outcomes in trastuzumab-refractory patients.

T-DXd was then evaluated in the second-line setting in Western patients who progressed on a trastuzumab-containing regimen in the single-arm DESTINY-Gastric02 with centrally confirmed HER2-positivity (n = 79) [39]. Significantly, this study mandated central screening to confirm HER2 status on tissue acquired after progression on trastuzumab-containing chemotherapy regimens. The primary endpoint was confirmed ORR by independent central review. Confirmed ORR was reported in 38% of patients recruited (95% CI 27.3–49.6), with completed responses documented in three patients. Median PFS was reported as 5.5 months (95% CI 4.2–7.3). The safety profile seen was consistent with the established toxicity profile seen in previous studies, and the majority of the adjudicated drug-related ILD/pneumonitis cases (n = 5/6) were grade ≤2. The phase III DESTINY-Gastric04 trial (NCT04704934) will ascertain if T-DXd is superior to second-line paclitaxel and ramucirumab in a global population and is currently recruiting.

The bystander effect associated with T-DXd could also establish a new subgroup of HER-low tumours, which could be treated with HER2-targeted agents. These patients have been previously treated as HER2-negative tumours. In the dose escalation component of the phase I study consisting of a population of heavily pre-treated gastric and breast cancers, T-DXd demonstrated anti-tumour activity even in low HER2-expressing tumours [40]. Subsequent evaluation of exploratory HER2-low cohorts, defined as patients with HER2 IHC 2+/ISH− and IHC 1+ disease, in the phase II DESTINY-Gastric01 study reported an ORR of 26.3% (n = 5/19) and 9.5% (n = 2/21) in these respective subgroups [41]. These results provide preliminary evidence of clinical activity in HER2-low patients who have been proven to be chemorefractory.

There may also be clinical utility in combining T-DXd with immune checkpoint inhibitors. In comparison to T-DXd alone, co-administration with anti-PD1 antibody therapy in a murine mouse model resulted in improved anti-tumour activity and prolonged OS [42]. Additionally, T-DXd upregulates tumour cell major histocompatibility complex (MHC) class I expression and dendritic cell activation markers. The results from the on-going phase Ib/II DESTINY-Gastric03 (NCT04379596) study will assess T-DXd monotherapy and, in combination with chemotherapy and/or pembrolizumab in a treatment-naïve population, will provide safety data on T-DXd-based combinations and preliminary efficacy data in gastric cancers.

#### 4.1.2. RC48

RC48 (disitamab vedotin) is comprised of hertuzumab, a humanised anti-HER2 antibody conjugated to the microtubule inhibitor monomethylauristatin E (MMAE) via a valine-citrulline linker. Preclinical assessment demonstrated that RC48 induced a stronger anti-tumour response in breast and gastric cancer cell lines, in addition to superior anti-tumour activity than trastuzumab emtansine in trastuzumab- and lapatinib-resistant xenograft tumour models [43]. A bystander effect was also observed with neighbouring HER2-negative cells in culture undergoing apoptosis upon recognition and internalisation of RC48 by HER2-overexpressing cells [44]. Subsequent phase I in 57 patients, of which 47 had gastric cancer, reported that RC48 was well-tolerated. The most frequent grade ≥3 treatment-related adverse events included myelosuppression, hypoesthesia and hyperbilirubinaemia [45]. A single arm phase II study was conducted with 125 Chinese patients with HER2-overexpressing gastric cancer who were refractory or intolerant to ≥2 lines of standard chemotherapy [46]. The majority of patients recruited (57.6%) had received previous trastuzumab. The independently assessed ORR was 24.8% and median PFS and OS were 4.1 months (95% CI 3.7–4.9 months) and 7.9 months (95% CI 6.7–9.9 months), respectively. Further phase III evaluation is on-going in a randomised study of comparing RC48 against physician’s choice chemotherapy or apatinib in HER2-positive gastric cancer patients in the chemorefractory setting (NCT04714190).

#### 4.1.3. SYD985

SYD985 (trastuzumab duocarmazine) is a HER2-targeting ADC comprising of trastuzumab covalently bound to a linker drug containing the DNA-alkylating agent duocarmycin, at a drug-to-antibody ratio of 2.8:1 [47]. SYD985 showed promising preclinical anti-tumour activity in solid tumour cell lines with varying HER2 expression, including those derived from gastric cancers. First-in-human assessment was conducted in patients with breast, gastric, urothelial and endometrial cancer with at least HER2 IHC expression of 1+ [48,49,50]. The non-breast cancer expansion cohort included 16 gastric adenocarcinoma patients. Of these, 1 patient had a partial response (ORR: 6%) and a median PFS was 3.2 months (95% CI 1.6–5.3 months) [51]. The safety, pharmacokinetics and efficacy of SYD985 is next being assessed in combination with niraparib, a PARP inhibitor, in HER2-expressing advanced solid tumours (NCT04235101) (Table 2).

### 4.2. HER2-Directed Immunotherapy

#### 4.2.1. Immune Checkpoint Inhibitors

Anti-programmed death-1 receptor (anti-PD-1) and anti-programmed death-ligand 1 (PD-L1) inhibitors are the most rigorously evaluated immune checkpoint inhibitors and have been found to be efficacious in various tumour types, including in subsets of gastric cancers [55,56,57,58]. In addition to inhibition of multiple pro-survival cellular signalling pathways, trastuzumab precipitates immunomodulation by increasing HER2 internalisation and cross presentation by dendritic cells to stimulate HER2-specific T cell responses [59,60,61,62]. Trastuzumab can also upregulate PD-1 and PD-L1 expression, induce expansion of tumour specific CD4 and CD8 T-cells and modulate MHC class II expression [63,64,65], which could enhance the efficacy of immune checkpoint inhibitors. The combination of trastuzumab with anti-PD1 therapy has demonstrated anti-tumour activity in a murine model [66], supporting its investigation in the clinic.

Proof-of-concept supporting the addition of pembrolizumab to capecitabine, platinum chemotherapy and trastuzumab was first demonstrated when a single-arm phase II study reported that 70% (n = 26/37) of patients treated with this combination were progression-free at 6-months [67]. An ORR of 91% (95% CI 78–97) and median PFS of 13.0 months (95% CI 8.6–not reached) was also reported and provided justification for a phase III, placebo-controlled investigation (KEYNOTE-811, NCT03615326). Results from a protocol-specified interim analysis from KEYNOTE-811 evaluated the ORR of the first 264 participants enrolled into the study [53]. Preliminary efficacy data generated from the interim analysis reported an ORR of 74.4% (95% CI 66.2–81.6) in patients who received pembrolizumab in comparison to 51.9% (95% CI 43.0–60.7) in the placebo group, with a greater number of complete responders in the pembrolizumab group (11.3% vs. 3.1%). The 22.7% improvement in ORR was statistically significant (95% CI 11.2–33.7; *p* = 0.00006), and similar rates of adverse events were seen between both arms. The U.S. Food and Drug Administration have granted accelerated approval to pembrolizumab in conjunction with trastuzumab and first-line chemotherapy in HER2-positive gastric cancer based on the results of this interim analysis. Although these results are promising, the primary analysis of KEYNOTE-811 will confirm if the addition of immune checkpoint inhibition to the current standard-of-care can significantly augment survival outcomes (Table 2).

The phase II INTEGA trial (NCT03409848) randomised 88 patients from 21 centres in Germany with previously untreated HER2-positive locally advanced or metastatic OG adenocarcinoma to ipilimumab or FOLFOX in combination with nivolumab and trastuzumab and posed the possibility of a chemotherapy-free option in this setting [68]. The primary endpoint was the OS rate at 12 months. At 12 months, the OS rate was 57% and 70% in the ipilimumab and FOLFOX arms, respectively. Median OS in the ipilimumab group was 16.4 months, whereas patients in the FOLFOX group has a median OS of 21.8 months. Analysis of outcomes including OS at 12 months, PFS, OS and ORR based on PD-L1 CPS positivity using thresholds of ≥1 and ≥5 did not show enhanced OS rates, implying that the effect of these combinations are independent of PD-L1 expression. This study also incorporated serial ctDNA assessment at baseline and following the first cycle of treatment. Patients who demonstrated clearance or stable levels of cfDNA after one cycle of treatment had a median OS of 31.2 months in comparison to 8.5 months in those who had an increase in cfDNA concentration of >20% (n = 21). The emergence of truncating and epitope-loss ERBB2 sequence variations were observed in four patients who developed trastuzumab resistance. Although the clinical outcomes from the patients randomised to FOLFOX were better than those seen with ipilimumab, the latter combination showed an OS rate at 12 months and median OS comparable to historical control data derived from the ToGA trial. These findings highlight the importance of chemotherapy backbone in the first-line treatment of advanced oesophago-gastric adenocarcinoma in the absence of biomarkers beyond HER2 and PD-L1 expression, as we have also seen with HER2-negative disease [69].

#### 4.2.2. Margetuximab

Margetuximab is an anti-HER2 monoclonal antibody engineered to bind selectively to stimulatory Fcγ receptor IIIA (CD16A) with simultaneous decreased binding to inhibitory Fcγ receptor IIB (CD32B) on natural killer cells. In vitro, margetuximab enhances tumour cell-directed ADCC, T-cell activity and upregulation of tumour cell PD-L1 expression, which collectively amplify both innate and adaptive immune responses. In a phase I study that recruited HER2-overexpressing solid tumour patients, 12% (n = 7/60) and 50% (n = 30/60) of patients achieved partial responses and stable disease with margetuximab [70]. Seventy percent of the recruited patients had disease refractory to previous HER2-targeted therapies. Ex vivo analyses of peripheral mononuclear cells isolated from trial participants showed that margetuximab augmented ADCC to a higher degree than trastuzumab.

Margetuximab, when assessed in combination with pembrolizumab in a phase Ib/II trial of 92 response-evaluable patients with HER2-positive gastric or GOJ cancers previously treated with trastuzumab, demonstrated preliminary anti-tumour activity with an ORR of 18% and a tolerable safety profile [71]. Objective response was most pronounced in biomarker-positive subgroups, particularly in patients with HER2 IHC3-positive/PD-L1 positive tumours, where an ORR of 44% was reported. The presence of *HER2* amplification in ctDNA was also associated with a better response rate when compared to patients with undetectable *HER2* amplification (ORR 15% vs. 2%). The median PFS of 2.73 months (95% CI 1.61–4.34) and median OS of 12.48 months (95% CI 9.07–14.09) was better than those seen in 10 previously published andomized studies assessing HER2-targeted therapies in the second-line setting (Table 1), providing proof-of-concept synergism between margetuximab and anti-PD1 blockade.

The phase II/III MAHOGANY study, which evaluates margetuximab with the anti-PD1 INCMGA00012 or the anti-LAG inhibitor retifanlimab as first-line therapy, is currently recruiting and will serve to move the investigational evaluation of margetuximab forward [54] (Table 2). Results from an interim analysis assessing efficacy and safety of the first 40 non-MSI-H patients treated with margetuximab and retifanlimab reported an ORR of 53% (95% CI 36–69%), including four confirmed complete responses and 17 partial responses [72]. The most commonly observed TRAEs included fatigue, infusion-related reaction, rash and diarrhoea. These results exceeded the prespecified futility boundary, and final results from this cohort will further inform clinicians on its role in HER2-targeted therapeutics in this tumour type.

#### 4.2.3. Zanidatamab (ZW25)

Zanidatamab is a bispecific antibody that targets two HER2 epitopes: extracellular domain 4, which contains the trastuzumab binding site, and extracellular domain 2, which contains the pertuzumab binding site. Preclinical observations describe a higher level of anti-tumour activity and more effective HER2 inhibition with ZW25 compared to trastuzumab and pertuzumab in combination [73]. Of the 33 patients with heavily pre-treated HER2-positive cancers who were recruited into the phase I study, 11 had OG cancer. An ORR of 43% and DCR of 56% was reported in the OG cohort, and there were no major safety concerns [74]. Results from a phase II study assessing zanidatamab will be assessed as first-line treatment in combination with chemotherapy (CAPOX, CF or mFOLFOX6); the study reported a confirmed ORR of 75% across all chemotherapy backbones and a median duration of response of 16.4 months [75]. Diarrhoea was the most frequent TRAE but was manageable in the outpatient setting and mitigated by prophylactic treatment. Zanidatamab will next be evaluated with chemo-immunotherapy as first-line treatment (HERIZON-GEA-01, NCT04276493) (Table 2).

#### 4.2.4. KN026

Another bispecific antibody that has shown preliminary activity in HER2-positive (IHC 3+/2+ ISH+) or HER2 low (IHC 1+/2+ ISH− or 0/1+ ISH+) GOJ and gastric adenocarcinoma patients is KN026. A total of 45 patients with HER2-positive disease refractory to at least one line of standard treatment were enrolled in a phase II study where they received KN026 monotherapy [76]. An ORR of 56% with duration of response (DoR) of 9.7 months was seen in the HER2-positive cohort. In contrast, the ORR in the HER2-low group was 14% and the DoR was 6.3 months. The most common treatment emergent adverse events observed with KN026 were liver function elevation, anaemia, rash and infusion-related reactions.

KN026 can also be safely combined with KN046, an anti-PD-L1 and anti-CTLA4 bispecific antibody, to engage both innate and adaptive immune responses. Results from the HER2-positive gastrointestinal tumour expansion cohort from a phase Ib study assessing KN026 and KN046 combination therapy showed an ORR of 86% (n = 6/7) in the first-line metastatic GOJ and gastric adenocarcinoma cohort, whereas the ORR in the later line cohort was 44% (n = 8/18) [77]. Grade ≥3 related AEs were neutropenia, thrombocytopenia, immune-mediated endocrinopathy, encephalitis, infusion-related reaction and pulmonary arterial hypertension (3.1% each). These results indicated that KN026 and KN046 is a safe combination with encouraging efficacy and provide a rationale for further investigation of this combination in HER2-positive OG adenocarcinoma.

#### 4.2.5. SBT6050

SBT6050 is comprised of a toll-like receptor 8 agonist (TLR8) linker payload conjugated to pertuzumab. SBT6050 was engineered to activate myeloid, natural killer (NK) and T cells in HER2-expressing tumours and has been shown to induce various antitumour immune mechanisms in preclinical studies [78]. Interim results from the dose escalation component of a first-in-human study of SBT6050 monotherapy (n = 14) and combination therapy (n = 4) with pembrolizumab [78] has reported a manageable safety profile with related treatment-emergent adverse events consistent with immune activation such as chills, diarrhoea and fatigue [78]. Common AEs were similar between mono- and combination therapy. The monotherapy dose of 0.6 mg/kg two-weekly was associated with detectable intratumoural SBT6050 payload, HER2 target saturation and pharmacodynamic markers indicative of myeloid, NK and T cell activation, and it was selected as the recommended phase 2 dose for further evaluation as a single agent and in combination with pembrolizumab for the expansion cohort. SBT6050 is also being evaluated in combination with either T-DXd or tucatinib and trastuzumab +/− capecitabine in HER2-positive breast, gastric, colorectal or non-small cell lung cancers [79] (Table 2).

#### 4.2.6. ALX 148

CD47 is a marker of self-recognition present on tumour and normal tissue that precipitates an anti-phagocytic signal upon binding to the signal regulatory protein alpha (SIRPα) transmembrane protein on the surface of macrophages [80]. CD47/SIRPα binding inhibits phagocytosis of healthy cells, whereas cells with low CD47 expression in damaged cells are susceptible to macrophage-mediated destruction. Tumour cells overexpress CD47 and therefore evade the macrophage component of immune surveillance. CD47 inhibition has also been shown to prime CD8 effector T cells with tumour specificity by increasing antigen processing and presentation by dendritic cells and tumour-associated macrophages in pre-clinical studies [81]. One potential disadvantage of exploiting CD47 as a therapeutic target is that CD47 is expressed on normal tissue as well, leading to undesirable on-target toxicity. For example, anaemia and thrombocytopenia were seen in animal studies with CD47 inhibitors with an active Fc domain [82].

ALX148 (evorpacept) is a CD47 myeloid checkpoint inhibitor that was engineered with an inactive Fc domain to minimise these haematological toxicities. ALX148 has been evaluated in the ASPEN-01 phase I clinical trial as monotherapy and as a component of combination therapies with chemotherapy and targeted agents. Within a gastric cancer cohort consisting of 38 patients with HER2-positive disease, 18 patients received ALX in combination with trastuzumab, ramucirumab and paclitaxel (ATRP), and a further 20 patients received ALX148 with trastuzumab [83]. Patients who received ATRP reported an ORR of 72% and 12-month OS rate of 76%, which compares favourably with results seen in the RAINBOW and DESTINY-01 studies. An ORR of 21.1% was reported in the ALX148 and trastuzumab cohort. Investigation into ALX148 as a therapeutic option in previously treated HER2-positive gastric cancers is currently being assessed in ASPEN-06. The phase II study aims to provide proof-of-concept of the effect of ALX148 in improving ORR, which was already achieved with trastuzumab, ramucirumab and paclitaxel, whereas the phase III will be a placebo-controlled study investigating ATRP against standard-of-care paclitaxel and ramucirumab (Table 2).

#### 4.2.7. Chimeric Antigen Receptor T (CAR-T) Cell Therapy

Chimeric antigen receptor T (CAR-T) cell therapy consists of engineered T cells isolated from a patient and modified using viral vectors to introduce the chimeric antigen receptor (CAR) to recognise specific tumour-associated antigens (TAA). The resultant CAR-T cells are amplified in vitro before being re-infused back into the patient, where T-cell proliferation, cytotoxicity and cytokine release precipitates targeted tumour cell destruction. As a surface antigen, HER2 is a putative CAR-T target. CAR-T cell therapy targeting the HER2 antigen on gastric cancer cells has shown cytotoxic activity against patient-derived HER2-positive gastric cancer cells and also demonstrated tumour inhibition and prolonged survival in xenograft models [84]. Several early phase clinical trials assessing the safety and preliminary clinical activity of CAR-T therapy in HER2-positive advanced solid tumours, including OG adenocarcinoma, are underway (Table 2).

#### 4.2.8. Vaccines

Various types of vaccine therapies are also in active development in HER2-positive cancers, albeit in phase I settings. Immunologic response analysis from a first-in-human study in eight patients with chemo-refractory HER2-expressing metastatic gastric cancer (IHC >1+) treated with BVAC-B, an autologous B cell- and monocyte-based immunotherapeutic vaccine transfected with recombinant *HER2*, showed that BVAC-B was capable of induction of NK and HER2-specific T-cells and release of HER2-specific antibody. The most common TRAE was fever, which was documented in 50% of patients, and cytokine release syndrome was documented in one patient, which was managed with supportive approaches. These results indicate that vaccine-based strategies are feasible in HER2-positive gastric cancers and have an acceptable safety profile [85]. On-going phase I studies include assessment of the HER2-directed vaccine TAEK-VAC-HerBy in combination with HER2-antibodies of immune checkpoint inhibitors in advanced solid tumours (NCT04660929), and a separate study assessing a peptide vaccine, which induces humoral immunity against HER2-expressing tumour cells, will provide further data on the clinical utility of immunotherapeutic vaccines in gastric cancers (NCT01417546) [86] (Table 2).

### 4.3. Tyrosine Kinase Inhibitors (TKI)

#### 4.3.1. Tucatinib

Tucatinib is a highly selective, small molecule TKI of HER2. Cellular signalling assays have shown that tucatinib is >1000-fold more selective for HER2 compared to the closely related EGFR, thus minimising the risk of EGFR-related toxicities that can be induced by dual HER2/EGFR inhibition. In addition to proven single-agent activity in vitro, tucatinib has also demonstrated clinical activity in murine models derived from N87 gastric cancer cell lines [87].

To date, clinical evaluation of tucatinib has largely been confined to HER2-positive breast and colorectal cancers. Phase III data has shown that tucatinib in combination with trastuzumab and capecitabine improves progression-free and overall survival in HER2-positive breast cancer patients previously treated with trastuzumab, pertuzumab and trastuzumab emtansine [88]. In a phase II study of 22 patients with chemorefractory colorectal cancers, an ORR of 55% was observed with tucatinib with trastuzumab combination therapy. In HER2-positive gastric cancers, the phase II/III trial MOUNTAINEER-02 is currently recruiting patients eligible for second-line systemic therapies who have previously been treated with a HER2-directed antibody (NCT04499924) [89]. The phase II component will evaluate the safety of combination therapy consisting of tucatinib, trastuzumab, paclitaxel and ramucirumab, followed by a placebo-controlled phase III evaluation of this combination therapy. An interesting aspect of the study is the incorporation of an exploratory cohort within the phase II component that will recruit patients who are HER2-negative on blood-based next generation sequencing but HER2-positive on baseline biopsy, which will provide insight into the correlation between HER2 alterations in blood and tissue in patients who have received prior HER2-directed therapy. Tucatinib is also being assessed in combination with trastuzumab and oxaliplatin-based chemotherapy in the first-line setting (NCT04430738) (Table 2).

#### 4.3.2. Afatinib

As an irreversible oral tyrosine kinase inhibitor that targets EGFR, HER2, ErbB3 and ErbB4 transphosphorylation, afatinib has the potential to circumvent trastuzumab resistance. The combination of afatinib and trastuzumab in a phase I study of 13 patients with HER2-positive advanced solid tumours reported an ORR of 7.7% and a clinical benefit rate of 69.2% [90]. At phase II level where 20 OG adenocarcinoma patients previously treated with trastuzumab received single-agent afatinib, an ORR of 10% (n = 2/20), median PFS of 2 months (95% CI 1.8–3.51) and median OS of 7 months (95% CI 3–11 months) was reported [33]. In a separate cohort of 12 patients who received afatinib and trastuzumab, the ORR was 8%. Afatinib monotherapy was well-tolerated and the most common TRAEs were diarrhoea and skin toxicity. Grade 2 diarrhoea requiring dose modifications despite prophylactic anti-diarrhoeal therapy was seen in 42% of patients who received combination therapy. *EGFR/ERBB2* coamplification and *MET* amplification were found to be associated with afatinib sensitivity and resistance, respectively [34]. The addition of afatinib to first-line cisplatin and 5-fluorouracil was not found to confer any clinical benefit in comparison to the ToGA regimen after a single-arm study of 55 patients with HER2-positive OG adenocarcinoma reported an ORR of 42.9%, median PFS of 5.0 months and median OS of 8.7 months [91]. Present investigation of afatinib focusses on combinatorial strategies with paclitaxel as second-line treatment in the advanced setting (NCT02501603, NCT01522768) (Table 2).

#### 4.3.3. Poziotinib

Poziotinib is an irreversible pan-HER TKI. Combination therapy comprising of pizotinib with paclitaxel and trastuzumab in advanced gastric cancer patients as second-line setting demonstrated a median PFS and OS of 13.0 weeks (95% CI 9.8–21.9) and 29.5 weeks (95% CI 17.9–59.2). The most common AEs deemed related to poziotinib were diarrhoea, rash, stomatitis, pruritus and anorexia. The results from a second phase I/II study of this combination are currently awaited (NCT01746771) (Table 2).

#### 4.3.4. Pyrotinib

Pyrotinib is a potent, irreversible TKI that inhibits signal transduction through the ERBB receptors. A phase I basket trial of pyrotinib in 62 patients with heavily pre-treated HER2-mutated or amplified advanced solid tumours showed an acceptable toxicity profile, with diarrhoea being the only grade ≥3 TRAE recorded in 24.2% of patients [92]. Encouraging efficacy data was demonstrated, with an ORR of 19% (95% CI 7–31%) and median PFS of 5.4 months (95% CI 4.4–7.3). Enhanced efficacy was seen when pyrotinib was used in conjunction with docetaxel (ORR 21%). Dysregulation of the CCND1-CDK4/6-Rb axis has been shown to contribute to pyrotinib resistance, which was reversed by the administration of the CDK4/6 inhibitor SHR6390 in murine models [93]. The combination of pyrotinib and SHR6390 resulted in an ORR of 60% in a small phase I trial consisting of five gastric cancer patients who failed standard systemic therapy, demonstrating proof-of-concept of this combination clinically [93].

## 5. HER2 Targeting in the Curative Paradigm

In contrast to the advanced setting where systemic therapy is guided by HER2 status, the benefit of targeting HER2 in the curative paradigm has yet to be defined. To date, the assessment of HER2-targeted approaches in early gastric cancer management is limited to relatively small studies. For example, a Spanish phase II trial consisting of 36 patients with HER2-positive gastric or GOJ adenocarcinoma has demonstrated the feasibility of adding trastuzumab to perioperative capecitabine and oxaliplatin chemotherapy and reported an 18-month DFS of 71% [94]. The single-arm HER-FLOT study evaluated trastuzumab in combination with FLOT chemotherapy using a primary endpoint of pathological complete response. Pathological complete response was seen in 21.5% (n = 12/56) of patients with locally advanced HER2-positive gastric cancers following neoadjuvant combination therapy, exceeding the pre-specified threshold of 20% to warrant further investigation [95]. Dual HER2-inhibition using trastuzumab and pertuzumab with FLOT chemotherapy in the randomised PETRARCA study (n = 81) has also been shown to significantly improve pathological complete response (35% vs. 12%, *p* = 0.02) and nodal response (68% vs. 39%) over FLOT alone [96]. However, this was at the expense of higher rates of grade ≥3 diarrhoea (41% vs. 5%) and leukopenia (23% vs. 13%) with combination treatment when compared to FLOT chemotherapy.

Further investigation into HER2-targeting in resectable gastric cancers is underway. In the peri-operative setting, the INNOVATION trial (NCT02205047) randomises HER2-positive patients to doublet fluoropyrimidine chemotherapy or FLOT or chemotherapy with trastuzumab and/or pertuzumab [97] (Table 2). Antibody-drug conjugates are also being assessed as neoadjuvant therapy in resectable gastric cancers; examples include the phase II EPOC2003 study, which will examine T-DXd in both HER2 overexpressed and HER2-low disease (NCT05034887) [52] and RC48 with chemo-immunotherapy in HER2-positive resectable gastric cancers (NCT05113459) (Table 2).

Adjuvant components of multimodal therapies in resectable gastric cancers are associated with low rates of commencement and further attrition in rates of completion, but there is a lack of data to inform the necessity of adjuvant therapy, particularly in patients with favourable features such as complete pathological response following pre-operative treatment. To address this unmet need, clinical trial design for the advancement of multimodal treatment strategies in gastric cancer will need to move towards a risk-stratified approach. One potential avenue is by exploiting the detection of minimal residual disease (MRD) following curative resection using ctDNA, which can identify patients at high risk of disease recurrence and is an adverse prognostic feature [98,99,100]. Additionally, detectable ctDNA following surgery can pre-empt radiological recurrence [101,102]. MRD detection has been incorporated into a pilot study of HER2-positive patients who have undergone standard-of-care surgery, neoadjuvant and adjuvant therapies and who have persistent ctDNA following curative resection and will be treated with adjuvant pembrolizumab and trastuzumab (NCT04510285) (Table 2). The primary endpoint will be rate of ctDNA clearance at 6 months.

## 6. Future Directions

Trastuzumab has anchored HER2-directed therapy in advanced gastro-oesophageal cancers for over a decade. After negative outcomes from successive phase III trials investigating the efficacy of alternative first- or second-line treatments in HER2-positive gastric cancers, more recent data has indicated efficacy with immunotherapy-based approaches and ADCs, and further insight into the biology of primary and acquired resistance to HER2-targeted agents has deepened our understanding into mechanisms of primary and acquired resistance.

Exploitation of the immune system against HER2-directed cells in gastric cancer is being evaluated with several approaches. At present, pembrolizumab with chemotherapy and trastuzumab holds an FDA approval for first-line treatment based on the 22.7% ORR improvement seen with the addition of pembrolizumab to chemotherapy and trastuzumab from the interim analysis of KEYNOTE-811 [53]. Although a final survival readout is required before this combination can be regarded as the new standard-of-care for first-line therapy, it holds promise, as this combination elicited comparatively deeper responses than chemotherapy and trastuzumab and appeared to be well-tolerated. An attractive aspect of immunotherapy is that these drugs are capable of inducing durable responses and long-term remissions, but the available data from KEYNOTE-811 is not mature enough to determine if this is the case with this particular regimen. It would also be interesting to see if PD-L1 expression can enrich for clinical benefit in HER2-positive gastric cancers, similar to what we have observed in HER2-negative patients where chemo-immunotherapy is already a standard-of-care for treatment-naive patients with a PD-L1 Combined Positive Score of ≥5. In KEYNOTE-811, patients were recruited irrespective of PD-L1 status, and 84.4% of the interim analysis population were PD-L1 CPS ≥1; a numerically greater difference in ORR was seen in these patients compared to those who were PD-L1 negative. Determining the effect of PD-L1 expression in relation to the efficacy of this combination would be informative, and it is hoped that further data will be published in due course. Other immunotherapy-based approaches such as margetuximab, which demonstrated favourable OS outcomes when used as second-line therapy with pembrolizumab in published second-line trials, and the bispecific antibody zanidatamab have also entered phase III evaluation [54,103]. Another immunotherapy-based strategy is CAR-T cell therapy, but developing CAR-T in solid tumours has been challenging on the whole due to factors such as tumour antigen heterogeneity and difficulties in CAR-T penetration into the tumour tissue through vascular endothelium [104]. Evaluation of HER2-directed CAR-T cell therapy is still at its early stages.

ADCs are a novel class of drug where its efficacy relies on target expression and subsequent internalisation of its cytotoxic payload. Its therapeutic potential in HER2-positive solid tumours is evident, as various iterations of this novel class of drug are being assessed at early phase level, and there are several clinical trials assessing HER2-directed ADCs specifically in gastric cancers (Table 2). T-DXd represents the first HER2-targeted drug that demonstrates efficacy in trastuzumab-refractory disease, both in the chemo-refractory setting in Asian patients and as second-line therapy in a Western population [38,39]. If results from the global, phase III DESTINY-Gastric04 are positive, T-DXd would be recognised as the gold standard second-line regimen for HER2-positive disease, following trastuzumab. However, its role as second-line treatment may be limited if KEYNOTE-811 reports positive survival results, leading to questions surrounding its clinical utility if first-line standard-of-care therapy evolves to include pembrolizumab. Earlier phase evaluation of T-DXd in the first-line and neoadjuvant settings are also on-going. Interstitial lung disease occurs in approximately 10% of patients who receive T-DXd and can lead to early cessation of this treatment. This may serve as a significant limitation to its use in early disease, as it could impact fitness for surgery, which is the key component of curative multimodality treatment. The bystander effect seen with T-DXd, deemed valuable in overcoming heterogenous HER2 expression, also has the potential to overhaul the treatment paradigm of advanced gastric cancers by introducing the concept of HER2-low disease. This provides the tantalising prospect of extending the therapeutic benefit of HER2-targeting to a wider population of patients. To develop a better understanding into the margin of benefit of T-DXd in HER2-low subgroups, future investigations should include comparisons of efficacy to not only HER2-positive patients treated with T-DXd but also to HER2-low patients who receive conventional therapy.

Meaningful strides in the progress of HER2-directed therapies can only be made if we understand mechanisms of response and resistance to these drugs. Recognising HER2 downregulation in response to trastuzumab, for example, has highlighted the value of repeat biopsy for re-assessment of HER2 status prior to initiation of HER2-directed second-line therapy in the context of a clinical trial. As newer efficacious therapies develop, so should our understanding of their mode of action in addition to response and resistance mechanisms. For instance, further pharmacokinetic analyses examining the relationship between target antigen expression levels, concentration of intratumoural payloads and antitumour efficacy of ADCs and comprehension of the bystander effect of ADCs is desirable [105]. The on-going clinical trials with integrated ctDNA assessment will also provide valuable data into longitudinal monitoring of driver and resistance mutations and will be integral to developing a more personalised treatment approach for these patients.

Although HER2-positive gastric cancers represent the minority of the overall patient population, it defines a subset of patients with an actionable molecular alteration—a rarity in the landscape of a tumour type where the development of targeted therapies has been disadvantaged by molecular heterogeneity. The multiple therapeutic agents in development that demonstrate potential in HER2-positive gastric cancer and are being developed in parallel signals an exciting era in the management of this disease. Critical evaluation of the clinical and translational research data from these trials will be central to advancing HER2-targeted therapies in gastric cancer and will hopefully lead to breakthroughs in the management of these patients.

## Figures and Tables

**Table 1 cancers-14-03824-t001:** Selected randomised HER2-targeted clinical trials in advanced oesophago-gastric adenocarcinoma. Adapted from [12].

Trial	Phase	HER2 Definition	Treatment Arms	*N*	Primary Endpoint	Results
**First-line therapy**						
ToGA [7]	III	IHC 3+ and/or ISH-positive	Capecitabine or 5-FU, cisplatin +/− trastuzumab	594	OS	mOS: 13.8 vs. 11.1 months (HR 0.74, 95% CI 0.60–0.91, *p* = 0.0046)
TRIO-013/LOGiC [10]	III	IHC3+ and/or ISH-positive	Capecitabine, oxaliplatin +/− lapatinib	545	OS	mOS: 12.2 vs. 10.5 months (HR 0.91, 95% CI 0.73–1.12, *p* = 0.32)
JACOB [8]	III	IHC 3+ or IHC 2+ ISH-positive	Capecitabine or 5-FU, cisplatin, trastuzumab +/− pertuzumab	780	OS	mOS: 17.5 vs. 14.2 months (HR 0.84, 95% CI 0.71–1.00, *p* = 0.057)
HELOISE [13]	IIIb	IHC 3+ or IHC 2+ ISH-positive	Cisplatin, capecitabine, trastuzumab 8 mg/kg loading dose + 6 mg/kg or 10 mg/kg maintenance dose	248	OS	mOS: 12.5 vs. 10.6 months (HR 1.24, 95% CI 0.86–1.78, *p* = 0.2401]
**Second-line therapy**						
TyTan [11]	III	ISH-positive	Paclitaxel +/− lapatinib	261	OS	mOS: 11.0 vs. 8.9 months (HR 0.84, 95% CI 0.64–1.11, *p* = 0.10)
GATSBY [9]	II/III	IHC 3+ or IHC 2+ ISH-positive	Trastuzumab emtansine vs. taxane	302	OS	mOS: 7.9 vs. 8.6 months (HR 1.15, 95% CI 0.87–1.51, *p* = 0.86)
T-ACT [14]	II	IHC 3+ or IHC 2+ ISH-positive	Paclitaxel +/− trastuzumab	91	PFS	mPFS: 3.2 vs. 3.7 months (HR 0.91, 80% CI 0.67–1.22, *p* = 0.33)
**Third-line therapy**						
DESTINY-Gastric01	II	IHC 3+ or IHC 2+ ISH-positive	Trastuzumab deruxtecan vs. physician’s choice chemotherapy (irinotecan or paclitaxel)	187	ORR	ORR: 42.8% (95% CI 33.8–52.3) vs. 12.3% (95% CI 5.2–24.1)

CI: confidence interval; HR: hazard ratio; IHC: immunohistochemistry; ISH: in-situ hybridisation; mOS: median overall survival; mPFS: median progression-free survival; ORR: overall response rate; OS: overall survival; PFS: progression-free survival; 5-FU: 5-fluorouracil.

**Table 2 cancers-14-03824-t002:** Selected HER2-targeted therapies undergoing clinical trial evaluation.

Investigational Medicinal Product	Clinical Trial	Phase	Intervention	Treatment Setting	Status	Primary Endpoint
**Antibody-drug conjugates**						
T-DXd	DESTINY-Gastric03 NCT04379596	I/II	Trastuzumab + platinum + chemotherapy vs. TDXd vs. TDXd + chemotherapy vs. T-DXd + chemotherapy + pembrolizumab vs. T-DXd + pembrolizumab	First-line	Recruiting	ORR
	DESTINY-Gastric04 NCT04704934	III	T-DXd vs. paclitaxel + ramucirumab	Second-line	Recruiting	OS
	EPOC2003 [52] NCT05034887	II	T-DXd	Neoadjuvant	Recruiting	Major pathological response
RC48	NCT04714190	III	RC48 vs. physician’s choice therapy (paclitaxel, irinotecan or apatinib)	Chemo-refractory	Recruiting	OS
	NCT05113459	II	RC48 + sintilimab + capecitabine	Neoadjuvant	Not yet recruiting	Pathologic complete response
SYD985	NCT04235101	I	SYD985 + niraparib	Chemo-refractory	Recruiting	Dose-limiting toxicity
**Immune checkpoint inhibitors**						
Pembrolizumab	NCT04510285	II	Trastuzumab + pembrolizumab in patients with minimal residual disease post-surgery	Adjuvant	Recruiting	Rate of ctDNA clearance at 6 months
	KEYNOTE-811 [53] NCT03615326	III	CAPOX/FOLFOX + trastuzumab + pembrolizumab/placebo	First-line	Recruitment completed	OS and PFS
**Margetuximab**						
Margetuximab	MAHOGANY [54] NCT04082364	II/III	Phase II: Margetuximab + retifanlimab Phase III: Trastuzumab vs. retifanlimab + margetuximab vs. tebotelimab + margetuximab + chemotherapy vs. megetuximab + chemotherapy	First-line	Recruiting	Phase II: ORR Phase III: OS
**Bispecific antibodies**						
Zanidatamab (ZW25)	HERIZON-GEA-01 NCT04276493	III	CAPOX/CF + trastuzumab vs. zanidatamab + CAPOX/CF vs. zanidatamab + tislelizumab + CAPOX/CF	First-line	Recruiting	OS and PFS
SBT6050	NCT04460456	I/Ib	SBT6050 monotherapy and SBT6050 + pembrolizumab	Chemo-refractory	Recruiting	Dose-limiting toxicity Adverse events ORR
	NCT05091528	I/II	SBT6050 + trastuzumab deruxtecan; or SBT6050 + trastuzumab + tucatinib + capecitabine; or SBT6040 + trastuzumab + tucatinib	Chemo-refractory	Recruiting	Dose-limiting toxicities Adverse events ORR
**CD47 inhibitors**						
ALX148	NCT05002127	II/III	Phase II: Trastuzumab + ramucirumab + paclitaxel +/− ALX148 Phase III: Trastuzumab + ramucirumab + paclitaxel + ALX148 vs. ramucirumab + paclitaxel + placebo + placebo	≥Second-line	Recruiting	Phase II: ORR Phase III: OS
**CAR-T cell therapy**						
BPX-603	NCT04650451	I	HER2-targeted dual-switch CAR-T cells	Chemo-refractory	Recruiting	Dose-limiting toxicities Maximum tolerated dose
CCT303-406	NCT04511871	I	CCT303-406	Chemo-refractory	Recruiting	Maximum tolerated dose
TACO1-HER2	NCT04727151	I/II	TACO1-HER2	Chemo-refractory	Recruiting	Adverse events
CAdVEC HER-specific CAR-T cells	NCT03740256	I	CAdVEC HER-specific CAR-T cells	Chemo-refractory	Recruiting	Dose-limiting toxicities
**Vaccines**						
TAEK-VAC-HerBy	NCT04246671	I/II	TAEK-VAC-HerBy	≥ Second-line	Recruiting	Dose-limiting toxicities
NHS-IL12	NCT01417546	I	Chimeric HER2 B-cell peptide vaccines	≥ Second-line	Recruitment completed	Dose-limiting toxicities Maximum tolerated dose
**Tyrosine kinase inhibitors**						
Tucatinib	MOUNTAINEER-02 NCT04430738	II/III	Tucatinib + trastuzumab + paclitaxel + ramucirumab vs. paclitaxel + ramucirumab	Second-line	Recruiting	Dose-limiting toxicities and adverse events (phase II) OS and PFS (phase III)
	NCT04499924	Ib/II	Tucatinib + trastuzumab + oxaliplatin-based chemotherapy	First-line	Recruiting	Dose-limiting toxicities Adverse events
Afatinib	NCT02501603	II	Afatinib + paclitaxel	Second-line	Active, not recruiting	PFS
	NCT01522768	II	Afatinib + paclitaxel	Second-line	Active, not recruiting	ORR
Poziotinib	NCT01746771	I/II	Poziotinib + paclitaxel + trastuzumab	Second-line	Recruitment completed	Dose-limiting toxicities Maximum tolerated dose

T-DXd: trastuzumab deruxtecan; ORR; overall response rate; OS: overall survival; PFS: progression-free survival.

## Data Availability

The data can be shared up on request.

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
