# Peer review of "HER2 Inhibition in Gastric Cancer—Novel Therapeutic Approaches for an Established Target"

_cancers, 2022, doi:10.3390/cancers14153824_

Round 1

Reviewer 1 Report

This review article highlights different strategies to target HER2-related cancers, their mechanism of action, how resistance arises for the HER2 therapies and ways to overcome them.

I find the article highly useful in learning about gastric cancers. The tables describing the different lines of treatment, their clinical efficacy, patient responses and ongoing clinical trials are informative.

I would encourage the authors to make an illustration to explain the mechanism of action for the different therapies to increase the usefulness of the paper.

I recommend this paper for publication. Congratulations to the authors.

Reviewer 2 Report

 Excellent! I appreciate the great effort that the authors have made to write this review article.

Reviewer 3 Report

The review of Fong and Chau is well written, and a contemporary update on recent developments for the tratemt of HER2-positive gastric cancers.

The authors may consider the following comments.

1) HER2 testing is notorously prone to sampling errors. When it comes to biopsies I advocate to refer to the recommendations set forth by hte American College of Pathologist, the Amercian Society of Clinical Pathology and ASCO (JCO 2017). A minimum of 5 tumor bearing biopsy specimens should be obtained for HER2-testing. Otherwise, HER2 status may reflect effect of sampling errors rather than treatment (Page 4).

2) The number of HER2-positive cases (according to ToGA criteria) eligible for treatment is still low. And despite recent advancements, the majority of gastric cancer patients will not benefit. This limitation merrits a notice in the conclusions.

3) However, having said this, ADCs bear the potential to increase patients eligible for HER2-targeted regimens (by including now also HER2 low cases). Therefore, the second sentence in the conlcusions may point out that the introduction of ADCs increases the numbers of patients eligible for targeted therapy, which I personally encounter one of the most important messages (apart form advancements made with ICIs).
